# Scrub typhus in pregnancy: A 10-year multicenter study in resource-limited settings in China

**Peilin Zhao[1], Tieyong Dong[2], Hongbin Lu[3], Rui Zhu[4], Shanshan Zhao[5], Wuqian Tao[6], Li Li[7], Chunmei Liu[8], Shuwei Pu[9], Ling Mo[10], Huanhuan Wang[11]\***

**1** Department of Neonatology, People's Hospital of Jinping Miao and Yao Dai Autonomous County, Honghe Hani and Yi Autonomous Prefecture, Yunnan Province, China, **2** Department of Neonatology, People's Hospital of Lvchun County, Honghe Hani and Yi Autonomous Prefecture, Yunnan Province, China, **3** Department of Infectious Disease, People's Hospital of Jinping Miao and Yao Dai Autonomous County, Honghe Hani and Yi Autonomous Prefecture, Yunnan Province, China, **4** Department of Neonatology, People's Hospital of Pingbian County, Honghe Hani and Yi Autonomous Prefecture, Yunnan Province, China, **5** Department of Neonatology, The First People's Hospital of Honghe State, Honghe Hani and Yi Autonomous Prefecture, Yunnan Province, China, **6** Department of Neonatology, People's Hospital of Hekou Yao Autonomous County, Honghe Hani and Yi Autonomous Prefecture, Yunnan Province, China, **7** Department of Neonatology, People's Hospital of Honghe County, Honghe Hani and Yi Autonomous Prefecture, Yunnan Province, China, **8** Department of Neonatology, People's Hospital of Yuanyang County, Honghe Hani and Yi Autonomous Prefecture, Yunnan Province, China, **9** Department of Infectious Disease Control, Honghe Prefecture Center for Disease Control and Prevention, Honghe Hani and Yi Autonomous Prefecture, Yunnan Province, China, **10** Department of Obstetrics and Gynecology, People's Hospital of Jinping Miao and Yao Dai Autonomous County, Honghe Hani and Yi Autonomous Prefecture, Yunnan Province, China, **11** Department of Neonatology, National Children's Medical Center/Children's Hospital of Fudan University, National Health Commission (NHC) Key Laboratory of Neonatal Diseases, Shanghai, China

\* huanhuan_wang@fudan.edu.cn

## Abstract

### Background

Despite its association with high mortality rates and negative fetal outcomes, large-scale epidemiological studies on scrub typhus (ST) during pregnancy remain scarce.

### Methods

We conducted a retrospective, multicenter study by collecting 260 pregnant women with ST in China across a 10-year time period to evaluate how clinical characteristics changed over this time and identify risk factors for poor fetal outcome.

### Results

In total, 78.5% (n = 204) presented with pathognomonic eschars or ulcers, and 75.4% (n = 196) had Weil-Felix test, in which 46.4% (n = 91) had a titer of ≥1:160 for OXK. A higher proportion of patients with pneumonia (18.7% vs. 9.5%, p = 0.033) and a longer length of hospitalization (7 vs. 6 days, p = 0.007) were seen in laboratory confirmed cases than clinically diagnosed cases. Compared to patients in the second trimester, a higher miscarriage rate (64.5% vs. 15.0%, p < 0.001) was observed in the first trimester, and a lower fetal death/stillbirth rate (1.0% vs 18.7%, p < 0.001) and a higher prematurity rate (21.0% vs.

**Data availability statement:** The data that support the findings of this study are provided as Supporting Information.

**Funding:** HW was supported by Key Development Program of Children's Hospital of Fudan University (Nos. EK2022ZX02 and EK2022PT06). The funders had no role in study design, data collection and analysis, decision to publish, or preparation of the manuscript.

**Competing interests:** The authors have declared that no competing interests exist.

9.3%, p = 0.019) were observed in the third trimester. The use of chloramphenicol (5.7% vs. 15.5%, p = 0.016) and fetal death/stillbirth rate (3.1% vs. 12.8%, p = 0.01) were decreased, but there was no significant improvement in fetal outcome in the second 5 years (35.1% vs. 39.7%, p = 0.466). Over 1/3 (n = 90) had a poor fetal outcome, including 40.0% (n = 36) miscarriage, 23.3% (n = 21) fetal death/stillbirth and 36.7% (n = 33) preterm birth. The use of macrolides reduced the risk of a poor fetal outcome, while lower gestational age (GA) at the time of infection, pneumonia, leucopoenia, thrombocytopenia, and use of chloramphenicol were associated with a poor fetal outcome in univariate logistic-regression analysis, but only lower GA at the time of infection and pneumonia were independent risk factors for a poor fetal outcome on multivariate logistic-regression analysis with an odds ratio of 6.6 (95% CI 2.4–17.8, p < 0.001) and 3.1 (95% CI 1.3–7.6, p = 0.011).

## Conclusions

This is the largest number of cases of ST in pregnancy so far. Our findings indicate that this population have a high risk of poor fetal outcome, especially those with lower GA at the time of infection and those with pneumonia at the same time. Further studies are needed to investigate the correlation between antibiotics regimens for this population and fetal outcome.

### Author summary

Scrub typhus (ST) is an acute febrile illness ranging in severity from mild to severe. Delayed diagnosis, lack of treatment, or improper management can lead to significant morbidity and mortality. Despite its association with high mortality rates and negative fetal outcomes, large-scale epidemiological studies on ST during pregnancy remain scarce. Less than 100 cases with pregnancy outcome reported from 1992 to 2014, and the reports after 2014 involved a small number of cases as well, in which the largest one analyzed 33 cases.

Our study includes the largest number of cases in this population to date. We investigated 260 pregnant women with ST in China across a 10-year time period. First, we observed that this population have a high risk of poor fetal outcome, especially those with lower gestational age at the time of infection and those with pneumonia at the same time. Second, the antibiotics regimens for this population have been changed in the second 5 years, but there was no significant improvement in fetal outcome. Further studies are needed to investigate the correlation between antibiotics regimens for this population and fetal outcome. Third, we found there is a lack of diagnostic tool for ST in resource-limited settings, policies and strategies should be strengthened to support the widespread use of accurate and affordable point-of-care testing for ST in the future.

## Introduction

Scrub typhus (ST), also known as tsutsugamushi disease, is caused by the arthropod borne gram-negative obligately intracellular bacillus *Orientia tsutsugamushi*. ST is a widely prevalent zoonotic disease endemic to the Asia-Pacific region, affecting about one million people each year [1–3].

ST is an acute febrile illness ranging in severity from mild to severe, with variable clinical manifestations ranging from mild self-limiting features like fever, headache, cough, and myalgia to fatal course complicated by acute respiratory distress syndrome, congestive heart failure, acute renal injury, septic shock, disseminated intravascular coagulation (DIC), and central nervous system dysfunction [4,5]. Delayed diagnosis, lack of treatment, or improper management can lead to significant morbidity and mortality. Without appropriate treatment, the mortality of ST can be up to 30–70%. The median mortality for untreated patients is 6%, while for treated patient is 1.4% [6,7].

ST can affect all age groups but only a few case reports and case series which focus on pregnant women were published in the past. Less than 100 cases with pregnancy outcome reported from 1992 to 2014 [8], and the reports after 2014 involved a small number of cases as well, in which the largest one analyzed 33 cases [9–14]. Due to the limited evidence in the literature regarding the impact of ST in pregnancy and some studies linking it to maternal death and adverse fetal outcomes such as miscarriage, intrauterine death, stillbirth, and preterm delivery [8–14], our study sought to exam clinical presentation, diagnosis, treatment, and outcome of pregnant women with ST over a 10-year period in Yunnan province in China to: 1) compare clinical characteristics and outcomes of clinically diagnosed cases and laboratory confirmed cases; 2) investigate the effects of the disease on patients in different stages of pregnancy; 3) compare patients from the first 5 years to the second 5 years to evaluate how clinical characteristics and outcomes changed over this time; 4) identify the risk factor for poor fetal outcome; 5) compare our results with latest literature published since 2014. This population could be valuable for future randomized, controlled interventional trials aimed at reducing morbidity and mortality, as well as improving fetal outcomes in pregnant women with ST, particularly in resource-limited settings.

## Methods

### Ethics statement

The Ethical Review Board of People's Hospital of Jinping Miao and Yao Dai autonomous county approved the study protocols [No.: (2024) LC-06] and individual consent for this retrospective analysis was waived.

### Study design and data collection

This is a multicenter, retrospective study that involved 7 hospitals in Yunnan province in China, including 2 level III and 5 level II hospitals. Coordination for this study was based at People's Hospital of Jinping Miao and Yao Dai autonomous county. We collected pregnant women with ST admitted to the 7 hospitals between January 1, 2014 and December 31, 2023. Data on maternal and fetal clinical characteristics, management, and outcomes among these patients were retrospectively collected from hospital records or through phone interviews.

ST is diagnosed based on epidemiologic exposure (travel to a disease-endemic area and contact with chiggers or rodents <3 weeks before the onset of illness), clinical manifestations (fever, lymphadenopathy, skin rash, and pathognomonic eschars or ulcers) and laboratory test: Weil-Felix test, indirect immunofluorescence antibody assay, polymerase chain reaction or isolation of *Orientia tsutsugamushi* [15–17]. Weil-Felix test is the only diagnostic tool available in our study. A minimum titer of 1:160 or a fourfold increase over previous levels for OXK is considered diagnostically significant [18,19].

Pregnancy is divided into three trimesters. First trimester spans from weeks 1 to 12 of pregnancy, second trimester covers weeks 13 to 27 of pregnancy, and third trimester is from week 28 until the birth. A miscarriage refers to the expulsion or removal of a fetus

(or embryo) weighing less than 500 grams, typically occurring around 22 weeks of gestation [20]. Fetal death refers to the death of a fetus before birth, regardless of the duration of pregnancy. Stillbirth occurs when a baby is born deceased after 22 weeks of completed pregnancy [21]. Preterm birth is defined as born before 37 weeks of gestational age (GA). Poor fetal outcomes refer to miscarriage, intrauterine fetal death, stillbirth and preterm birth.

## Statistical Analysis

The data were analyzed using SPSS software (v. 26.0, SPSS Inc., Chicago, IL). Continuous variables were presented as median and interquartile range (IQR) and categorical data were expressed as number and percentage (%). Chi square or Fisher's exact test were used for categorical variables, and Mann-Whitney U test was applied for comparing continuous variables. Pearson's correlation was used for GA at the time of infection and fetal outcome. The risk of poor fetal outcome was analyzed by uni- and multivariate logistic-regression analysis, and odds ratio (OR) and 95% confidence interval (CI) was calculated around the point estimation value. A P value <0.05 was considered as statistically significant.

## Results

A total of 260 pregnant women were diagnosed with ST. The clinical presentation, diagnosis, treatment, and outcome were compared for patients among the clinically diagnosed cases and laboratory confirmed cases, the different stages of pregnancy, patients in the 2 time periods (2014–2018, 2019–2023), and patients with good and poor fetal outcome. We also compare our findings with the latest literature published since 2014.

### Clinically diagnosed cases versus laboratory confirmed cases

In total, 78.5% (n = 204) presented with pathognomonic eschars or ulcers, and 75.4% (n = 196) had Weil-Felix test, in which 52.5% (n = 103), 1.0% (n = 2) and 46.4% (n = 91) had a titer of 1:40, 1:80, and ≥1:160 for OXK, respectively.

There were 65.0% (n = 169) clinically diagnosed cases and 35.0% (n = 91) laboratory confirmed cases (Table 1). The median (IQR) time from the onset of fever to the definitive diagnosis of ST was 4 (3–5) days. Once ST was diagnosed, antibiotic therapy for ST was commenced. There were no statistically significant differences in antibiotic treatment and fetal outcome, but a higher proportion of patients with pneumonia (18.7% vs. 9.5%, p = 0.033) and a longer length of hospitalization (LOS) (7 vs. 6 days, p = 0.007) were seen in laboratory confirmed cases than clinically diagnosed cases.

### Patients in different stages of pregnancy

There were 15.0% (n = 39), 46.5% (n = 121) and 38.5% (n = 100) patients who developed the disease in their first, second and third trimester of pregnancy, respectively (Table 2). There were no significant differences in clinical presentation and major complications in patients in any trimester, while patients in the first trimester had the highest proportion of leucopoenia (35.9% vs.9.1%, P < 0.001; 35.9% vs. 3.0%, p < 0.001) and poor fetal outcome (71.0% vs. 43.0% vs. 22.0%, p < 0.01), and the least use of macrolides (66.7% vs. 79.3%, P < 0.001; 66.7% vs. 92.0%, p < 0.001), while the use of chloramphenicol was least in patients in the third trimester (5.0% vs. 13.2%, p = 0.009; 5.0% vs. 23.1%, p < 0.01). The miscarriage rate of patients in the first trimester was significantly higher than it was in the second trimester (64.5% vs. 15.0%, p < 0.001). Compared to the patients in the second trimester, ones in the third trimester had a lower proportion of fetal death/stillbirth (1.0% vs 18.7%, p < 0.001), and a higher proportion of preterm birth (21.0% vs. 9.3%, p = 0.019).

**Table 1. Clinically diagnosed cases versus laboratory confirmed cases: clinical profile, management and outcome of 260 pregnant women with ST.**

| Diagnosis | Total | Clinically diagnosed | Laboratory confirmed | P value |
|---|---|---|---|---|
| N of patients, n (%) | 260 | 169 (65.0) | 91 (35.0) | – |
| Pregnancy in weeks | 24 (18–30) | 24 (18–32) | 24 (19–28) | 0.471 |
| First trimester, n (%) | 39 (15.0) | 24 (14.2) | 15 (16.5) | 0.623 |
| Second trimester, n (%) | 121 (46.5) | 76 (45.0) | 45 (49.5) | 0.490 |
| Third trimester, n (%) | 100 (38.5) | 69 (40.8) | 31 (34.1) | 0.285 |
| T ≥ 39°C, n (%) | 165 (63.5) | 113 (66.9) | 52 (57.1) | 0.235 |
| Duration of fever, days | 6 (4–8) | 6 (5–8) | 6 (4–8) | 0.453 |
| Eschar, n (%) | 204 (78.5) | 169 (100) | 35 (38.5) | <0.001 |
| Lymphadenopathy, n (%) | 10 (3.8) | 5 (3.0) | 5 (5.5) | 0.499 |
| Skin rash, n (%) | 7 (2.7) | 4 (2.4) | 3 (3.3) | 0.968 |
| Hepatomegaly, n (%) | 1 (0.4) | 1 (0.6) | 0 | 1.000 |
| Splenomegaly, n (%) | 7 (2.7) | 6 (3.6) | 1 (1.1) | 0.445 |
| Electrolyte imbalances, n (%) | 173 (66.5) | 114 (67.5) | 59 (64.8) | 0.669 |
| Hypoproteinemia, n (%) | 95 (36.5) | 58 (34.3) | 37 (40.7) | 0.369 |
| Hepatic dysfunction, n (%) | 105 (40.4) | 74 (43.8) | 31 (34.1) | 0.128 |
| Renal dysfunction, n (%) | 35 (13.5) | 19 (11.2) | 16 (17.6) | 0.139 |
| Pneumonia, n (%) | 33 (12.7) | 16 (9.5) | 17 (18.7) | 0.033 |
| Patients did Weil-Felix test, n (%) | 196 (75.4) | 105 (62.1) | 91 (100) | <0.001 |
| Leucopoenia, n (%) | 28 (10.8) | 21 (12.4) | 7 (7.7)) | 0.245 |
| Thrombocytopenia, n (%) | 65 (25.0) | 49 (29.0) | 16 (17.6) | 0.053 |
| Anemia, n (%) | 138 (53.1) | 90 (53.3) | 48 (52.7) | 0.917 |
| Use of macrolides, n (%) | 214 (82.3) | 139 (82.2) | 75 (82.4) | 0.973 |
| Use of chloramphenicol, n (%) | 30 (11.5) | 17 (10.1) | 13 (14.3) | 0.309 |
| Use of tetracycline antibiotics, n (%) | 16 (6.2) | 13 (7.7) | 3 (3.3) | 0.160 |
| Duration of antibiotic, days | 5 (5–7) | 5 (5–7) | 5 (5–7) | 0.175 |
| LOS, days | 6 (5–8) | 6 (5–8) | 7 (6–8) | 0.007 |
| Cost, CNY, $10^3$ | 2.6 (1.9–3.8) | 2.9 (1.7–3.8) | 2.6 (1.8–3.7) | 0.767 |
| Known fetal outcome, n (%) | 238 (91.5) | 158 (93.5) | 80 (87.9) | 0.123 |
| Total fetal loss, n (%) | 57 (23.9) | 35 (22.2) | 22 (27.5) | 0.361 |
| Miscarriage, n (%) | 36 (15.1) | 23 (14.6) | 13 (16.3) | 0.731 |
| Fetal death/Stillbirth, n (%) | 21 (8.8) | 12 (7.6) | 9 (11.3) | 0.348 |
| Preterm birth, n (%) | 33 (13.9) | 18 (11.4) | 15 (18.8) | 0.121 |
| Poor fetal outcome, n (%) | 90 (37.8) | 53 (33.5) | 37 (46.3) | 0.056 |

Abbreviations: ST, scrub typhus; T, temperature; LOS, length of hospitalization; CNY, Chinese Yuan (7CNY = 1USD). Values are given in n (%), or median and interquartile range (IQR). Chi square or Fisher's exact test were used for categorical variables, and Mann-Whitney U test was applied for comparisons of continuous variables.

### 2014–2018 versus 2019–2023

Compared to the patients in 2014–2018, patients in 2019–2023 presented with a higher proportion of an eschar (84.8% vs. 74.2%, p = 0.042), splenomegaly (5.7% vs. 0.6%, p = 0.013), and electrolyte imbalances (75.2% vs. 60.6%, p = 0.014), and had a higher proportion of macrolides use (89.5% vs. 77.4%, p = 0.012), but the proportion of renal dysfunction (5.7% vs. 18.7%, p = 0.002), chloramphenicol use (5.7% vs. 15.5%, p = 0.016) and fetal death/stillbirth (3.1% vs. 12.8%, p = 0.01) were significantly decreased. Although the positive rate of Weil-Felix test was relatively lower (33.3% vs. 56.9%, p = 0.001), more patients did Weil-Felix test in 2019–2023 (82.9% vs. 72.3%, p = 0.021) (Table 3).

**Table 2. Clinical profile, diagnosis, management and outcome of 260 pregnant women with ST in different stages of pregnancy.**

| Trimester | First | Second | Third | P value |
|---|---|---|---|---|
| N of patients, n (%) | 39 (15.0) | 121 (46.5) | 100 (38.5) | – |
| Pregnancy in weeks | 10 (7–12) | 21 (19–24) | 32 (29–36) | <0.001 |
| T ≥ 39°C, n (%) | 25 (64.1) | 82 (67.8) | 58 (58.0) | 0.243 |
| Duration of fever, days | 8 (5–9) | 6 (4–8) | 6 (4–8) | 0.105 |
| ≤ 3 days, n (%) | 5 (12.8) | 16 (13.2) | 21 (21.0) | 0.253 |
| 4–6 days, n (%) | 12 (30.8) | 50 (41.3) | 36 (36.0) | 0.423 |
| ≥ 7 days, n (%) | 22 (56.4) | 54 (44.6) | 43 (43.0) | 0.348 |
| Eschar, n (%) | 32 (82.1) | 95 (78.5) | 77 (77.0) | 0.809 |
| Lymphadenopathy, n (%) | 2 (5.1) | 5 (4.1) | 3 (3.0) | 0.681 |
| Skin rash, n (%) | 0 (0) | 3 (2.5) | 4 (4.0) | 0.671 |
| Hepatomegaly, n (%) | 0 (0) | 1 (0.8) | 0 (0) | 1.000 |
| Splenomegaly, n (%) | 1 (2.6) | 5 (4.1) | 1 (1.0) | 0.321 |
| Patients did Weil-Felix test, n (%) | 26 (66.7) | 97 (80.2) | 73 (73.0) | 0.193 |
| Positive Weil-Felix test, n (%) | 15 (57.7) | 45 (46.4) | 31 (42.4) | 0.559 |
| Leucopoenia, n (%)* | 14 (35.9) | 11 (9.1) | 3 (3.0) | <0.001 |
| Thrombocytopenia, n (%) | 15 (38.5) | 28 (23.1) | 22 (22.0) | 0.111 |
| Anemia, n (%) | 14 (35.9) | 66 (54.5) | 58 (58.0) | 0.058 |
| Electrolyte imbalances, n (%) | 29 (74.4) | 81 (66.9) | 63 (63.0) | 0.428 |
| Hypoproteinemia, n (%) | 15 (38.5) | 42 (34.7) | 38 (38.0) | 0.896 |
| Hepatic dysfunction, n (%) | 20 (51.3) | 51 (42.1) | 34 (34.0) | 0.152 |
| Renal dysfunction, n (%) | 8 (20.5) | 13 (10.7) | 14 (14.0) | 0.373 |
| Pneumonia, n (%) | 6 (15.4) | 14 (11.6) | 13 (13.0) | 0.818 |
| Myocarditis, n (%) | 1 (2.6) | 2 (1.7) | 1 (1.0) | 0.652 |
| Heart failure, n (%) | 0 (0) | 1 (0.8) | 1 (1.0) | 1.000 |
| DIC, n (%) | 1 (2.6) | 1 (0.8) | 0 (0) | 0.278 |
| Septic shock, n (%) | 0 (0) | 2 (1.7) | 0 (0) | 0.641 |
| Pleural effusion, n (%) | 0 (0) | 0 (0) | 2 (2.0) | 0.641 |
| Use of macrolides, n (%)* | 26 (66.7) | 96 (79.3) | 92 (92.0) | 0.001 |
| Use of chloramphenicol, n (%)** | 9 (23.1) | 16 (13.2) | 5 (5.0) | 0.008 |
| Use of tetracycline antibiotics, n (%) | 4 (10.2) | 9 (7.4) | 3 (3.0) | 0.201 |
| Duration of antibiotic, days | 5 (5–7) | 5 (5–6) | 5 (5–7) | 0.744 |
| LOS, days | 7 (5–8) | 6 (5–8) | 6 (5–8) | 0.791 |
| Cost, CNY, 10³ | 3.1 (2.0–4.5) | 2.7 (1.9–3.8) | 2.5 (1.8–3.8) | 0.393 |
| Maternal mortality, n (%) | 0 (0) | 0 (0) | 0 (0) | – |
| Poor -fetal outcomes, n (%)*** | 22 (71.0) | 46 (43.0) | 22 (22.0) | <0.001 |
| Miscarriage, n (%)**** | 20 (64.5) | 16 (15.0) | 0 (0) | <0.001 |
| Fetal death/Stillbirth, n (%)***** | 0 (0) | 20 (18.7) | 1 (1.0) | <0.001 |
| Preterm birth, n (%)****** | 2 (6.5) | 10 (9.3) | 21 (21.0) | 0.033 |

Abbreviations: ST, scrub typhus; T, temperature; DIC, disseminated intravascular coagulation; LOS, length of hospitalization; CNY, Chinese Yuan (7CNY = 1USD).
Values are given in n (%), or median and interquartile range (IQR). Chi square or Fisher's exact test were used for categorical variables, and Mann-Whitney U test was applied for comparisons of continuous variables.

*First vs. Second, P < 0.001; First vs. Third, p < 0.001.

**First vs. Third, p < 0.001; Second vs. Third, p = 0.009.

***First vs. Second, P = 0.006; First vs. Third, p < 0.001; Second vs. Third, p = 0.001.

****First vs. Second, P < 0.001.

*****Second vs. Third, p < 0.001.

******Second vs. Third, p = 0.019.

**Table 3. 2014–2018 Versus 2019–2023: clinical profile, diagnosis, management and outcome of 260 pregnant women with ST.**

| Year | 2014–2018 | 2019–2023 | P value |
|---|---|---|---|
| N of patients, n (%) | 155 (59.6) | 105 (40.4) | – |
| Pregnancy in weeks | 24 (19–30) | 24 (18–30) | 0.330 |
| First trimester, n (%) | 23 (14.8) | 16 (15.2) | 0.929 |
| Second trimester, n (%) | 70 (45.2) | 51 (48.6) | 0.589 |
| Third trimester, n (%) | 62 (40.0) | 38 (36.2) | 0.536 |
| T ≥ 39°C, n (%) | 104 (67.1) | 61 (58.1) | 0.243 |
| Duration of fever, days | 6 (4–8) | 6 (4–8) | 0.973 |
| Eschar, n (%) | 115 (74.2) | 89 (84.8) | 0.042 |
| Lymphadenopathy, n (%) | 5 (3.2) | 5 (4.8) | 0.762 |
| Skin rash, n (%) | 5 (3.2) | 2 (1.9) | 0.798 |
| Hepatomegaly, n (%) | 1 (0.6) | 0 (0) | 1.000 |
| Splenomegaly, n (%) | 1 (0.6) | 6 (5.7) | 0.013 |
| Electrolyte imbalances, n (%) | 94 (60.6) | 79 (75.2) | 0.014 |
| Hypoproteinemia, n (%) | 60 (38.7) | 35 (33.3) | 0.436 |
| Hepatic dysfunction, n (%) | 59 (38.1) | 46 (43.8) | 0.354 |
| Renal dysfunction, n (%) | 29 (18.7) | 6 (5.7) | 0.002 |
| Pneumonia, n (%) | 19 (12.3) | 14 (13.3) | 0.798 |
| Patients did Weil-Felix test, n (%) | 109 (70.3) | 87 (82.9) | 0.021 |
| Positive Weil-Felix test, n (%) | 62 (56.9) | 29 (33.3) | 0.001 |
| Leucopoenia, n (%) | 12 (7.7) | 16 (15.2) | 0.061 |
| Thrombocytopenia, n (%) | 42 (27.1) | 23 (21.9) | 0.349 |
| Anemia, n (%) | 86 (55.5) | 52 (49.5) | 0.356 |
| Use of macrolides, n (%) | 120 (77.4) | 94 (89.5) | 0.012 |
| Use of chloramphenicol, n (%) | 24 (15.5) | 6 (5.7) | 0.016 |
| Use of tetracycline antibiotics, n (%) | 11 (7.1) | 5 (4.8) | 0.442 |
| Duration of antibiotic, days | 5 (5–7) | 5 (5–6) | 0.554 |
| LOS, days | 7 (5–8) | 6 (5–8) | 0.199 |
| Cost, CNY, $10^3$ | 2.3 (1.7–3.1) | 3.4 (2.5–4.9) | <0.001 |
| Total fetal loss, n (%) | 38 (27.0) | 19 (19.6) | 0.082 |
| Miscarriage, n (%) | 20 (14.2) | 16 (16.5) | 0.625 |
| Fetal death/Stillbirth, n (%) | 18 (12.8) | 3 (3.1) | 0.010 |
| Preterm birth, n (%) | 18 (12.8) | 15 (15.5) | 0.554 |
| Poor fetal outcome, n (%) | 56 (39.7) | 34 (35.1) | 0.466 |

Abbreviations: ST, scrub typhus; LOS, length of hospitalization; CNY, Chinese Yuan (7CNY = 1USD). Values are given in n (%) or median and interquartile range (IQR). Chi square or Fisher's exact test were used for categorical variables, and Mann-Whitney U test was applied for comparisons of continuous variables.

## Maternal outcome

There was no maternal death due to the infection, but 3.1% (n = 8) admitted to ICU, including 1 complicated with pneumonia and heart failure, 1 with pneumonia and pleural effusion, 1 with pneumonia and septic shock, 1 with septic shock, 1 with pleural effusion, 1 with heart failure and 2 with DIC, in which 2 received chloramphenicol.

## Good versus poor fetal outcome

The fetal outcome of 8.5% (n = 22) patients were unknown. For the remaining 238 patients, 37.8% (n = 90) had poor fetal outcome, including 40.0% (n = 36) miscarriage, 23.3% (n = 21) fetal death/stillbirth and 36.7% (n = 33) preterm birth.

Compared to patients with good fetal outcome, the proportion of poor fetal outcome was highest in patients in the first trimester (71.0% vs. 22.0%, P < 0.001). The GA at the time of infection was strongly correlated with fetal outcomes, with the risk of poor outcomes decreasing as the gestational age advanced (r = -0.783, P < 0.001). Patients with poor fetal outcome had a higher proportion of pneumonia (20.0% vs. 8.1%, p = 0.007), leucopoenia (18.9% vs. 5.4%, p = 0.001), thrombocytopenia (31.1% vs. 19.6%, p = 0.038), and chloramphenicol use (20.0% vs. 5.4%, p < 0.001), but had a lower proportion of macrolides use (71.1% vs. 90.5%, p < 0.001) (Table 4).

Lower GA at the time of infection, pneumonia, use of chloramphenicol, leucopoenia and thrombocytopenia were associated with a poor fetal outcome, while the use of macrolides reduced the risk of a poor fetal outcome in univariate logistic-regression analysis with p < 0.05. However, only lower GA at the time of infection and pneumonia were independent risk factors for a poor fetal outcome on multivariate logistic-regression analysis with an odds ratio of 6.6 (95% CI 2.4–17.8, p < 0.001) and 3.1 (95% CI 1.3–7.6, p = 0.011) (Table 5).

**Table 4. Good Versus poor fetal outcome: clinical profile, diagnosis, management of 238 pregnant women with ST.**

| Fetal outcome | Total | Good | Poor | P value |
|---|---|---|---|---|
| N | 238 | 148 (62.2) | 90 (37.8) | – |
| First trimester, n (%) | 31 (13.0) | 9 (29.0) | 22 (71.0) | <0.001 |
| Second trimester, n (%) | 107 (45.0) | 61 (57.0) | 46 (43.0) | 0.137 |
| Third trimester, n (%) | 100 (42.0) | 78 (78.0) | 22 (22.0) | <0.001 |
| T ≥ 39°C, n (%) | 149 (62.6) | 93 (62.8) | 56 (62.2) | 0.938 |
| Duration of fever, days | 6 (4–8) | 6 (4–8) | 6 (5–9) | 0.486 |
| Eschar, n (%) | 189 (79.4) | 118 (79.7) | 71 (78.9) | 0.876 |
| Lymphadenopathy, n (%) | 9 (3.8) | 3 (2.0) | 6 (6.7) | 0.086 |
| Skin rash, n (%) | 6 (2.5) | 5 (3.4) | 1 (1.1) | 0.413 |
| Hepatomegaly, n (%) | 1 (0.4) | 1 (0.7) | 0 | 1.000 |
| Splenomegaly, n (%) | 7 (2.9) | 5 (3.4) | 2 (2.2) | 0.713 |
| Electrolyte imbalances, n (%) | 154 (64.7) | 89 (60.1) | 65 (72.2) | 0.058 |
| Hypoproteinemia, n (%) | 86 (36.1) | 51 (34.5) | 35 (38.9) | 0.521 |
| Hepatic dysfunction, n (%) | 96 (40.3) | 54 (36.5) | 42 (46.7) | 0.121 |
| Renal dysfunction, n (%) | 31 (13.0) | 15 (10.1) | 16 (17.8) | 0.119 |
| Pneumonia, n (%) | 30 (12.6) | 12 (8.1) | 18 (20.0) | 0.007 |
| Patients did Weil-Felix test, n (%) | 178 (74.8) | 109 (73.6) | 69 (76.7) | 0.603 |
| Positive Weil-Felix test, n (%) | 80 (44.9) | 43 (39.4) | 37 (53.6) | 0.064 |
| Leucopoenia, n (%) | 25 (10.5) | 8 (5.4) | 17 (18.9) | 0.001 |
| Thrombocytopenia, n (%) | 57 (23.9) | 29 (19.6) | 28 (31.1) | 0.038 |
| Anemia, n (%) | 129 (54.2) | 82 (55.4) | 47 (52.2) | 0.698 |
| Use of macrolides, n (%) | 198 (83.2) | 134 (90.5) | 64 (71.1) | <0.001 |
| Use of chloramphenicol, n (%) | 26 (10.9) | 8 (5.4) | 18 (20.0) | <0.001 |
| Use of tetracycline antibiotics, n (%) | 14 (5.9) | 6 (4.1) | 8 (8.9) | 0.124 |
| Duration of antibiotic, days | 5 (5–7) | 5 (5–7) | 5 (5–7) | 0.548 |
| LOS, days | 6 (5–8) | 6 (5–8) | 7 (5–9) | 0.199 |
| Cost, CNY, $10^3$ | 2.6 (1.9–3.8) | 2.6 (1.9–3.8) | 2.6 (1.9–3.8) | 0.735 |

Abbreviations: ST, scrub typhus; LOS, length of hospitalization; CNY, Chinese Yuan (7CNY = 1USD). Values are given in n (%) or median and interquartile range (IQR). Chi square or Fisher's exact test were used for categorical variables, and Mann-Whitney U test was applied for comparisons of continuous variables.

Table 5. **Risk factors for poor fetal outcome of 238 pregnant women with ST by uni-and multivariate logistic regression analysis.**

| Variables | Univariate | | Multivariate | |
|---|---|---|---|---|
| | OR (95% CI) P value | | OR (95% CI) P value | |
| Stage of pregnancy | – | <0.001 | – | – |
| Third trimester | 1 (reference) | – | 1 (reference) | – |
| Second trimester | 2.674 (1.455–4.914) | 0.002 | 2.206 (1.149–4.237) | 0.017 |
| First trimester | 8.667 (3.494–21.498) | <0.001 | 6.550 (2.406–17.835) | <0.001 |
| Pneumonia | 2.833 (1.293–6.208) | 0.009 | 3.144 (1.229–7.611) | 0.011 |
| Use of chloramphenicol | 4.375 (1.815–10.548) | 0.001 | 1.959 (0.457–8.394) | 0.356 |
| Use of macrolides | 0.257 (0.126–0.526) | <0.001 | 0.528 (0.158–1.762) | 0.299 |
| Leucopoenia | 4.190 (1.725–10.178) | 0.002 | 2.287 (0.835–6.267) | 0.108 |
| Thrombocytopenia | 1.884 (1.030–3.446) | 0.040 | 1.014 (0.490–2.099) | 0.971 |

Abbreviations: ST, scrub typhus; OR, odds ratio; CI, confidence interval.

## Our report versus latest literature after 2014

There were 7 cohorts and case series (cases ≥5) for ST in pregnancy since 2014, where 6 studies are from India and 1 is from Thailand [8–14]. The comparison between our results and the 7 available studies is shown in Table 6.

Totally, there were 398 cases of pregnant women with ST, in which 13.6% (n = 54), 40.8% (n = 158) and 44.7% (n = 173) patients who developed the disease in their first, second and third trimester of pregnancy, respectively.

Six reports found fever in all the patients. The proportion of patients with an eschar was >18.0% in 5 reports while none of the patients had an eschar in 3 reports. The proportion of patients with lymphadenopathy, skin rash, hepatomegaly or splenomegaly varied greatly, and it was <5% in total. In 4 reports, hepatic dysfunction and anemia were observed in >80% and >70% patients, respectively. Five studies reported the proportion of thrombocytopenia was > 20%.

The maternal mortality was 1.3% (n = 5), and poor fetal outcome was observed in 40.9% (n = 153) patients, including 64.0% (n = 98) fetal loss and 36.0% (n = 55) preterm birth. The rate of fetal death/still birth ranged from 8.8% to 20.0%. Three reports showed the miscarriage rate was >20.0% and fetal loss rate in the first trimester was highest.

## Discussion

ST can affect all age groups but has been rarely reported among pregnant women [8–14]. We reported a multicenter study in clinical presentation, diagnosis, treatment, and outcome among 260 pregnant women with ST over a 10-year period in China. To the best of our knowledge, it's the largest number of cases investigated so far.

Our study found that this population have a high risk of poor fetal outcome, especially for those who develop the disease in the first trimester. Fetal outcome is a major concern in pregnant women with infection. Pathogens, such as 'TORCH' (*Toxoplasma gondii*, other, rubella virus, cytomegalovirus, herpes simplex virus), Escherichia coli, group B streptococcus and enterococcus species, can be vertically transmitted to the fetus leading to congenital anomalies in 3% of live births and 10–30% of all stillbirths [22–23]. Although there was no report showing congenital anomalies due to vertically transmitted *Orientia tsutsugamushi*, ST was associated with poor fetal outcomes, including miscarriage, intrauterine death, stillbirth, and preterm delivery [8–14]. Yadav et al [10] reported poor fetal outcome in the form of

**Table 6. Our report versus those reported in the latest literature since 2014.**

| | Our report | Bahadur et al (2024) [9] | Yadav et al (2023) [10] | Kumar et al (2016) [11] | Rajan et al (2016) [12] | Meena et al (2016) [13] | McGready et al (2014) [8] | Sengupta et al (2014) [14] | Total* (data available) |
|---|---|---|---|---|---|---|---|---|---|
| Publication type | Cohort | case series | Cohort | case series | Cohort | case series | Cohort | Cohort | |
| Country | China | India | India | India | India | India | Thailand | India | |
| Cases, n | 260 | 5 | 27 | 14 | 33 | 6 | 11 | 42 | 398 |
| Laboratory confirmed, n (%) | 91 (35.0) | 5 (100) | 17 (63.0) | NA | NA | 6 (100) | 11 (100) | 42 (100) | 172 (49.0) |
| Fever, n (%) | 260 (100) | 5 (100) | 27 (100) | 14 (100) | NA | 6 (100) | 11 (100) | NA | 323 (100) |
| Eschar, n (%) | 204 (78.5) | 0 | 0 | 3 (21.4) | 17 (51.5) | 0 | 2 (18.2) | 15 (35.7) | 241 (60.6) |
| Lymphadenopathy, n (%) | 10 (3.8) | NA | 0 | 4 (28.6) | NA | 1 (16.7) | NA | NA | 15 (4.9) |
| Skin rash, n (%) | 7 (2.7) | NA | 0 | 2 (14.3) | NA | 0 | 2 (18.2) | NA | 11 (3.5) |
| Hepatomegaly, n (%) | 1 (0.4) | NA | 2 (7.4) | NA | 4 (12.0) | 1 (16.7) | 3 (27.3) | NA | 11 (3.3) |
| Splenomegaly, n (%) | 7 (2.7) | NA | 1 (3.7) | NA | 4 (12.0) | 1 (16.7) | 2 (18.2) | NA | 15 (4.5) |
| Hepatic dysfunction, n (%) | 105 (40.4) | 4 (80.0) | 24 (88.9) | 12 (92.3) | NA | 6 (100) | NA | NA | 151 (48.4) |
| Renal dysfunction, n (%) | 35 (13.5) | 0 | 2 (7.4) | 5 (35.7) | NA | 2 (33.3) | NA | NA | 44 (14.1) |
| Leucopoenia, n (%) | 28 (10.8) | 0 | 17 (63.0) | 0 | NA | 0 | NA | NA | 45 (14.4) |
| Thrombocytopenia, n (%) | 65 (25.0) | 2 (40.0) | 8 (29.6) | 3 (21.4) | NA | 6 (100) | NA | NA | 84 (26.9) |
| Anemia, n (%) | 138 (53.1) | 5 (100) | 20 (74.1) | 11 (78.6) | NA | 6 (100) | NA | NA | 180 (57.7) |
| Antibiotics in use | Macrolides Tetracycline Chloramphenicol | Azithromycin | Azithromycin Doxycycline | Azithromycin | Azithromycin Doxycycline | Azithromycin Doxycycline | NA | NA | |
| First trimester, n (%) | 39 (15.0) | 0 | 1 (3.7) | 0 | 5 (15.2) | 0 | 3 (27.3) | 6 (14.3) | 54 (13.6) |
| Second trimester, n (%) | 121 (46.5) | 1 (20.0) | 2 (7.4) | 8 | 9 (27.3) | 1 (16.7) | NA | 16 (38.1) | 158 (40.8) |
| Third trimester, n (%) | 100 (38.5) | 4 (80.0) | 20 (74.1) | 6 | 19 (57.6) | 4 (66.7) | NA | 20 (47.6) | 173 (44.7) |
| Post-partum, n (%) | 0 | 0 | 4 (14.8) | 0 | 0 | 1 (16.7) | 0 | 0 | 5 (1.3) |
| N of known fetal outcome, n (%) | 238 (91.5) | 5 (100) | 27 (100) | 14 (100) | 32 (97.0) | 6 (100) | 10 (90.9) | 42 (100) | 374 (94.0) |
| Fetal loss in first trimester, n (%) | 20 (64.5) | 0 | 1 (100) | 0 | 4 (80.0) | NA | NA | 4 (66.7) | 29 (56.9) |
| Fetal loss in second trimester, n (%) | 36 (33.6) | 1 | NA | 0 | 2 (22.2) | 1 (100) | NA | 6 (37.5) | 46 (29.5) |
| Fetal loss in third trimester, n (%) | 1 (1.0) | 0 | NA | 0 | 8 (42.1) | 0 | NA | 4 (20.0) | 13 (7.5) |
| Miscarriage, n (%) | 36 (15.1) | 0 | 1 (3.7) | 0 | NA | 2 (33.3) | 2 (20.0) | 10 (23.8) | 51 (14.9) |
| Fetal death/Still-birth, n (%) | 21 (8.8) | 1 (20.0) | 4 (14.8) | 0 | NA | 1 (16.7) | 2 (20.0) | 4 (9.5) | 33 (9.6) |
| Total fetal loss, n (%) | 57 (23.9) | 1 (20.0) | 5 (18.5) | 0 | 14 (42.4) | 3 (50.0) | 4 (40.0) | 14 (33.0) | 98 (26.2) |
| Preterm birth, n (%) | 33 (13.9) | 2 (40.0) | 12 (44.4) | 0 | 3 (9.1) | 3 (50.0) | 2 (20.0) | 0 | 55 (14.7) |
| Poor fetal outcome | 90 (37.8) | 3 (60.0) | 17 (63.0) | 0 | 17 (53.1) | 6 (100) | 6 (60.0) | 14 (33.3) | 153 (40.9) |
| Maternal mortality, n (%) | 0 | 1 (20.0) | 1 (3.7) | 0 | 1 (3.0) | 1 (16.7) | 0 | 1 (2.4) | 5 (1.3) |

Abbreviations: NA, not available.

intrauterine death (7.4%), miscarriage (3.7%), and stillbirths (3.7%) in 27 cases. McGready et al [8] reviewed clinical profiles of 97 pregnant women with ST or murine typhus, in which 17.3% (14/81 cases) had miscarriage and 41.8% (28/67 cases) had poor fetal outcomes (stillbirth, low birth weight, and preterm delivery). Sengupta et al [14] analyzed 42 cases of pregnant women with ST and revealed that pregnancy loss with ST was significantly higher as compared to their routine obstetric data (33% vs 2.8%, P < 0.001). In our study, over 1/3 had poor fetal outcomes in the form of miscarriage, fetal death/stillbirth and preterm delivery. Notably, over the 10-year time period, we observed lower GA at the time of infection was an independent risk factors for a poor fetal outcome on multivariate logistic-regression analysis with an odds ratio of 6.6 (95% CI 2.4–17.8, p < 0.001), and patients with ST in the first trimester experienced the worst fetal outcomes, with a miscarriage rate of 64.5%, significantly higher than the rate observed in all recognized pregnancies (15.3%) [20]. The fetal loss rate in the first, second and third trimester was 64.5%, 33.6% and 1.0%. The risk of poor fetal outcome decreased with advancing GA, which was in line with the reports from Rajan et al [12] and Sengupta et al [14]. The elevated rate of poor fetal outcomes in the first trimester may be linked to the high usage of chloramphenicol, which was associated with adverse fetal outcomes in our study's univariate logistic regression analysis. Although some studies have indicated that chloramphenicol use during the first trimester is not associated with teratogenicity [24,25], these studies primarily addressed topical or oral use. In contrast, chloramphenicol was administered intravenously to patients with ST in our study. Given the side effects of chloramphenicol, such as fatal aplastic anemia, bone marrow suppression, neurotoxicity, severe metabolic acidosis and gray baby syndrome, chloramphenicol is classed as a pregnancy category C drug and should be avoided in pregnancy [26]. Notably, the use of chloramphenicol decreased significantly, and the proportion of poor fetal outcomes was relatively lower in the latter half of our observation period. However, the direction of this association remains unclear. In addition, our study found the use of macrolides reduced the risk for poor fetal outcome in univariate logistic-regression analysis. The use of macrolides increased significantly in the latter half of our observation period, but there was no significant change in the proportion of poor fetal outcomes. Although macrolides are among the most commonly prescribed antibiotics during pregnancy, a cohort study from UK [27] showed macrolides in the first trimester of pregnancy were associated with an increased risk of any major malformation (OR1.55, 95% CI: 1.19–2.03), and specifically cardiovascular malformations (OR 1.62, 95% CI:1.05–2.51) compared with penicillin antibiotics, and macrolides in any trimester were associated with an increased risk of genital malformations (OR1.58, 95% CI: 1.14-2.19). In contrast, Andersson et al [28] found that the use of macrolides during pregnancy is not associated with an increased risk of major birth defects. Unfortunately, the management of ST remains controversial, and no specific treatment regimen has demonstrated a clear advantage or disadvantage in terms of efficacy or safety [29,30], not to mention the recommendation for pregnant women with ST. Further studies are needed to identify the most appropriate antibiotics for this population.

One limitation of this study is the use of the Weil-Felix test for laboratory diagnosis. This test is less preferred due to its lack of specificity and sensitivity, as well as the delay in antibody production. Therefore, diagnosis is mainly based on clinical manifestations and epidemiologic exposure in our study. In resource-limited settings, future policies and strategies should focus on supporting the widespread use of accurate and affordable point-of-care testing for ST. Another limitation of this study is its retrospective observational design, which means that differences in staffing and clinical practice across various hospitals may impact clinical outcomes in these patients. Hence, the results in our report may not be entirely consistent or directly comparable with data from published clinical trials due to limitations in diagnostic methods and variations in the severity of patients. Despite its limitations, this retrospective

multicenter study advances our understanding and could provide valuable insights for future randomized, controlled interventional trials aimed at reducing morbidity and mortality and improving fetal outcomes for pregnant women with ST in resource-limited settings.

## Conclusions

This retrospective multicenter study provides a real-world analysis of pregnant women with ST management in resource-limited settings in China. To the best of our knowledge, this study includes the largest number of cases in this population to date. Our findings indicate that patients with lower GA at the time of infection and pneumonia might be more prone to have poor fetal outcome. Further studies are needed to investigate the correlation between antibiotics regimens for this population and fetal outcome. Additionally, policies and strategies should be strengthened to support the widespread use of accurate and affordable point-of-care testing for ST in resource-limited settings.

## Supporting information

**S1 Data. The raw data of our study were shown in this file, which included clinical presentation, diagnosis, treatment, complications, and fetal outcomes among 260 pregnant women with scrub typhus.**
(XLSX)

## Acknowledgments

The authors thank Dr. Zhiguo Zhou from the University of Kansas Medical Center, Kansas City, Kansas, USA for embellishing and editing this manuscript.

## Author contributions

**Conceptualization:** Huanhuan Wang.

**Data curation:** Peilin Zhao, Tieyong Dong, Hongbin Lu, Rui Zhu, Shanshan Zhao, Wuqian Tao, Li Li, Chunmei Liu, Shuwei Pu, Ling Mo, Huanhuan Wang.

**Investigation:** Peilin Zhao, Tieyong Dong, Hongbin Lu, Rui Zhu, Shanshan Zhao, Wuqian Tao, Li Li, Chunmei Liu, Shuwei Pu, Ling Mo, Huanhuan Wang.

**Methodology:** Huanhuan Wang.

**Supervision:** Huanhuan Wang.

**Writing – original draft:** Peilin Zhao, Tieyong Dong, Hongbin Lu, Rui Zhu, Shanshan Zhao, Wuqian Tao, Li Li, Chunmei Liu, Shuwei Pu, Ling Mo, Huanhuan Wang.

**Writing – review & editing:** Huanhuan Wang.

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
