## [Decision Letter · Decision Letter 0]

23 Dec 2024

PNTD-D-24-01393

Scrub typhus in pregnancy: A 10-year multicenter study in resource-limited settings in China

Dear Dr. Wang,

Thank you for submitting your manuscript to PLOS Neglected Tropical Diseases. After careful consideration, we feel that it has merit but does not fully meet PLOS Neglected Tropical Diseases's publication criteria as it currently stands. Therefore, we invite you to submit a revised version of the manuscript that addresses the points raised during the review process.

Please submit your revised manuscript within 60 days Feb 21 2025 11:59PM. If you will need more time than this to complete your revisions, please reply to this message or contact the journal office at plosntds@plos.org. Please include the following items when submitting your revised manuscript:

We look forward to receiving your revised manuscript.

Kind regards,

Craig W. Roberts

Guest Editor

Mathieu Picardeau

Section Editor

Shaden Kamhawi

co-Editor-in-Chief

Paul Brindley

co-Editor-in-Chief

**Additional Editor Comments :**

Please address all reviewer comments in your revised manuscript.

**Journal Requirements:**

2) We note that your Data Availability Statement is currently as follows: "The authors are unable to share any pertinent data at the time of editorial and peer review." Please confirm at this time whether or not your submission contains all raw data required to replicate the results of your study. Authors must share the “minimal data set” for their submission. PLOS defines the minimal data set to consist of the data required to replicate all study findings reported in the article, as well as related metadata and methods (https://journals.plos.org/plosone/s/data-availability#loc-minimal-data-set-definition ).

3) Please amend your detailed Financial Disclosure statement. This is published with the article. It must therefore be completed in full sentences and contain the exact wording you wish to be published.

**Reviewers' Comments:**

Reviewer's Responses to Questions

**Key Review Criteria Required for Acceptance?**

**Methods**

-Are the objectives of the study clearly articulated with a clear testable hypothesis stated?

-Is the study design appropriate to address the stated objectives?

-Is the population clearly described and appropriate for the hypothesis being tested?

-Is the sample size sufficient to ensure adequate power to address the hypothesis being tested?

-Were correct statistical analysis used to support conclusions?

-Are there concerns about ethical or regulatory requirements being met?

Reviewer #1: Objective clear , study design- appropriate ,statastics correctly used

Reviewer #2: 1. Please provide the day of illness on which treatment was begun.

2. A single titer of 1:80 of Proteus OXK agglutination is not specific. What is the incidence of titer of 1:80 in healthy persons in Yunnan? As recognized by the authors, Weil Felix serology is not specific. A fourfold rise or greater in titer would be supportive of the diagnosis for purposes of this study. According to the diagnostic criteria, exposure to a disease endemic area, rash, and a titer of 1:80 would be confirmatory of the diagnosis. This is not sufficient. Many diseases could meet these criteria.

3. Table 1 what does "duration of fever, dates" mean? Does this reflect the entire course of illness or the number of days prior to treatment?

4. Tables 1, 2, and 3: What is the difference between "Weil-Felix test" and "positive Weil-Felix test"?

5. Lines 411 and 412: Please cite a reference documenting congenital anomalies due to vertically transmitted Orientia tsutsugamushi. I do not believe that this occurs.

**Results**

-Does the analysis presented match the analysis plan?

-Are the results clearly and completely presented?

-Are the figures (Tables, Images) of sufficient quality for clarity?

Reviewer #1: All matched

Reviewer #2: The results should be separated into groups that are documented by seroconversion or by only a single Weil-Felix titer and the presence of an eschar

The study contains a robust number of cases, the great majority of which manifested eschar and an interesting analysis of the fetal outcomes by trimester.

**Conclusions**

-Are the conclusions supported by the data presented?

-Are the limitations of analysis clearly described?

-Do the authors discuss how these data can be helpful to advance our understanding of the topic under study?

-Is public health relevance addressed?

Reviewer #1: Please add ICU / hdu admissions also.ST with pneumonia frequently need icu admissions. Add this important finding also .could not understand the rationale of using chloramphenicol in pregnancy, please elaborate . Thanks

Reviewer #2: The conclusions are difficult to assess owing to the inadequate confirmation of the diagnosis of scrub typhus.

**Editorial and Data Presentation Modifications?**

Reviewer #1: Minor revision.

Reviewer #2: See below

**Summary and General Comments**

Reviewer #1: Please add ICU / hdu admissions also.ST with pneumonia frequently need icu admissions. Add this important finding also .could not understand the rationale of using chloramphenicol in pregnancy, please elaborate . Thanks

Reviewer #2: If the authors stratified their results according to confirmed cases vs cases based upon the clinical diagnosis only, this might be a useful contribution of new knowledge.

PLOS authors have the option to publish the peer review history of their article (what does this mean? ). If published, this will include your full peer review and any attached files.

**Do you want your identity to be public for this peer review?** For information about this choice, including consent withdrawal, please see our Privacy Policy .

Reviewer #1: **Yes: ** Minakshi Rohilla

Reviewer #2: No

**Figure resubmission:**

While revising your submission, please upload your figure files to the Preflight Analysis and Conversion Engine (PACE) digital diagnostic tool, https://pacev2.apexcovantage.com/ . PACE helps ensure that figures meet PLOS requirements. To use PACE, you must first register as a user. Registration is free. Then, login and navigate to the UPLOAD tab, where you will find detailed instructions on how to use the tool. If you encounter any issues or have any questions when using PACE, please email PLOS at figures@plos.org. Please note that Supporting Information files do not need this step. If there are other versions of figure files still present in your submission file inventory at resubmission, please replace them with the PACE-processed versions.
---

## [Editor Report · Decision Letter 1]

8 Jan 2025

Dear Doctor Wang,

We are pleased to inform you that your manuscript 'Scrub typhus in pregnancy: A 10-year multicenter study in resource-limited settings in China' has been provisionally accepted for publication in PLOS Neglected Tropical Diseases.

Best regards,

Craig W. Roberts

Guest Editor

Mathieu Picardeau

Section Editor

Shaden Kamhawi

co-Editor-in-Chief

Paul Brindley

co-Editor-in-Chief

---

## [Editor Report · Acceptance letter]

Dear Doctor Wang,

We are delighted to inform you that your manuscript, "Scrub typhus in pregnancy: A 10-year multicenter study in resource-limited settings in China," has been formally accepted for publication in PLOS Neglected Tropical Diseases.

Best regards,

Shaden Kamhawi

co-Editor-in-Chief

Paul Brindley

co-Editor-in-Chief
